# The HslV Protease from *Leishmania major* and Its Activation by C-terminal HslU Peptides

**DOI:** 10.3390/ijms20051021

**Published:** 2019-02-26

**Authors:** Ndeye Mathy Kebe, Krishnananda Samanta, Priyanka Singh, Joséphine Lai-Kee-Him, Viviana Apicella, Nadine Payrot, Noémie Lauraire, Baptiste Legrand, Vincent Lisowski, Diane-Ethna Mbang-Benet, Michel Pages, Patrick Bastien, Andrey V. Kajava, Patrick Bron, Jean-François Hernandez, Olivier Coux

**Affiliations:** 1Centre de Recherche en Biologie cellulaire de Montpellier (CRBM), CNRS, Université de Montpellier, 34090 Montpellier, France; mathykebe@hotmail.fr (N.M.K.); noemie.lauraire@gmail.com (N.L.); andrey.kajava@crbm.cnrs.fr (A.V.K.); 2Institut des Biomolécules Max Mousseron (IBMM), CNRS, Université de Montpellier, ENSCM, Faculté de Pharmacie, 34090 Montpellier, France; krishnanandasamanta5@gmail.com (K.S.); singh.priyanka021@gmail.com (P.S.); vivianaapicella@libero.it (V.A.); nadine.payrot@laposte.net (N.P.); baptiste.legrand@umontpellier.fr (B.L.); vincent.lisowski@umontpellier.fr (V.L.); 3Centre de Biochimie Structurale (CBS), INSERM, CNRS, Université de Montpellier, 34090 Montpellier, France; josephine.laikeehim@cbs.cnrs.fr (J.L.-K.-H.); patrick.bron@cbs.cnrs.fr (P.B.); 4PIBBS, Biocampus Montpellier, CNRS, INSERM, Université de Montpellier, 34000 Montpellier, France; 5Laboratory of Parasitology-Mycology, MIVEGEC, CNRS, IRD, Université de Montpellier, 34000 Montpellier, France; diane-ethna.benet@umontpellier.fr (D.-E.M.-B.); pagesmichel@orange.fr (M.P.); patrick.bastien@umontpellier.fr (P.B.); 6Institut de Biologie Computationnelle, Université de Montpellier, 34000 Montpellier, France

**Keywords:** ATP-dependent protease, HslVU, peptide chemical synthesis, cryo-electron microscopy, Leishmania, mitochondria, allosteric regulation, cyclic peptides

## Abstract

HslVU is an ATP-dependent proteolytic complex present in certain bacteria and in the mitochondrion of some primordial eukaryotes, including deadly parasites such as *Leishmania*. It is formed by the dodecameric protease HslV and the hexameric ATPase HslU, which binds via the C-terminal end of its subunits to HslV and activates it by a yet unclear allosteric mechanism. We undertook the characterization of HslV from *Leishmania major* (LmHslV), a trypanosomatid that expresses two isoforms for HslU, LmHslU1 and LmHslU2. Using a novel and sensitive peptide substrate, we found that LmHslV can be activated by peptides derived from the C-termini of both LmHslU1 and LmHslU2. Truncations, Ala- and D-scans of the C-terminal dodecapeptide of LmHslU2 (LmC12-U2) showed that five out of the six C-terminal residues of LmHslU2 are essential for binding to and activating HslV. Peptide cyclisation with a lactam bridge allowed shortening of the peptide without loss of potency. Finally, we found that dodecapeptides derived from HslU of other parasites and bacteria are able to activate LmHslV with similar or even higher efficiency. Importantly, using electron microscopy approaches, we observed that the activation of LmHslV was accompanied by a large conformational remodeling, which represents a yet unidentified layer of control of HslV activation.

## 1. Introduction

ATP-dependent proteases are present in all living organisms [1] and play major roles in intracellular proteolysis, and therefore, in cell homeostasis. They rid the cells of old and misfolded proteins and are also involved in the control, often tightly regulated, of the stability of most regulatory proteins. In eukaryotes, the major ATP-dependent protease is the 26S proteasome responsible for the degradation of poly-ubiquitylated proteins. This multi-enzymatic complex is composed of the 20S catalytic core that is capped at one or both ends by its regulatory subunit, the 19S complex or PA700 [2,3]. In bacteria, several ATP-dependent proteases are present. Among them, HslVU, also called ClpYQ, is a proteolytic complex initially described in *Escherichia coli* (*E. coli*) [4,5,6]. It is a self-compartmentalized protease constituted of two HslV hexameric rings that enclose a proteolytic chamber containing 12 active sites whose catalytic residues are the N-terminal Thr (Thr1) of each subunit. HslV activity is controlled by its regulator HslU, an ATPase of the AAA^+^ family [7]. HslU forms a hexameric ring that binds in the presence of ATP to one or both ends of the protease, resulting into the formation of the HslVU ATP-dependent protease. Substrate recognition is most likely performed by the HslU regulator that then unfolds and translocates the captured protein into the internal proteolytic chamber of HslV, by an ATP-dependent mechanism [6,8,9,10].

HslV alone can only slowly degrade some hydrophobic peptides and unfolded proteins in vitro whereas it can be activated several folded by the binding of HslU [5,6]. Crystallographic and mutational analyses have shown that the C-terminus segment of HslU that is buried between the HslU subunits when the complex is at rest, distends in the presence of ATP and inserts into pockets located between adjacent HslV subunits [8]. Binding of HslU results into an enlargement of HslV pores [11,12], which most likely facilitates the entrance of substrates into the catalytic cavity of HslV. However, HslV activation additionally requires conformational rearrangements at or close to the active sites of HslV, which are propagated over a relatively long distance from the binding pockets to the active sites, allowing the latter to adopt an active configuration by shifting from an “off” to an “on” state [8,9,13,14,15].

Importantly, it has been shown that synthetic peptides composed of the 8-12 last amino acid residues of HslU can partially or fully replace the latter for HslV activation toward small fluorogenic peptide substrates or unfolded proteins, but not folded proteins [15]**.**

Whereas HslU is an ATPase that has no obvious similarity outside the AAA ATPase module with the ATPase subunits of the 26S proteasome, HslV is clearly related to 20S proteasome subunits, as shown by sequence similarities, enzymatic mechanism and structural organization [4,5,16,17]. Thus, HslVU was initially considered as an ancestral form of the proteasome that had evolved in eukaryotes into the 26S proteasome, making the coexistence of both proteases in the same living organism unlikely. However, while HslVU is indeed absent in higher eukaryotes, it has been discovered in the mitochondrion of trypanosomatids [18,19]. Furthermore, it has been shown that HslV and HslU genes are in fact present in most eukaryotic lineages [20]. Interestingly, from the studies on HslVU functions in these eukaryotes, it appears that, in contrast to its bacterial counterpart [21], the protease HslVU is essential for the survival of several parasites [22,23,24,25,26]. Indeed, it has been shown that both HslV and HslU1/U2 have important roles in mitochondrial DNA replication and cell cycle control in *Trypanosoma brucei* [22,26], and in transcription of genes encoded by the mitochondrial genome as well as cellular and mitochondrial growth in *Plasmodium falciparum* [23,25]. Thus, thanks to its essential functions in these dangerous parasites and its absence in humans, HslVU represents an attractive potential drug target to fight against deadly parasitic diseases.

A clear route for inhibition of HslV is to develop compounds targeting its active site(s), as done extensively for the eukaryotic 20S proteasome, for example [27]. However, although previous studies show the feasibility of species-specific inhibitors of proteasome [28,29,30], developing selective HslV inhibitors not targeting the human proteasome remains a challenge. Another approach to inhibit HslVU could be to target the formation of the HslVU complex, since association of the HslV and HslU subcomplexes is necessary for protein degradation by the protease. Such inhibition could be achieved in principle by small molecules [31] or high affinity mimetics of the C-terminal segment of HslU, which should prevent the docking of HslU to HslV by occupying its insertion pockets on HslV. Although such compounds could still activate HslV catalysis, they should nevertheless prevent the degradation of protein substrates that depends on HslUs, and thus, severely impair parasite growth. In fact, the validity of this approach has been already documented in *P. falciparum* [23]. In line with this idea, and to pave the way for the future development of specific compounds inhibiting the binding of HslU to HslV, we undertook an exploration of the structure-activity relationships of HslV-HslU interaction in *Leishmania major*.

Although the eubacterial HslVU protease has been extensively studied, much less is known about its eukaryote counterparts from a biochemical and structural point of view: the most detailed analysis to date is the characterization of *T. brucei* HslV [32,33]. Similar to *T. brucei*, *L. major*, which is the causative agent of cutaneous leishmaniasis, expresses together with HslV (LmHslV, UniProt ID Q4Q116) two isoforms of HslU, named LmHslU1 and LmHslU2 (UniProt IDs Q4QFH5 and Q4QI03, respectively). Our recent studies suggest that these two isoforms have specific roles [26], but whether they function as hetero- (U1/U2) or homo- (U1 and U2) complexes is presently not known.

In this study, we present the initial characterization of the proteolytic core LmHslV and its activation by peptides derived from the C-termini of HslUs. Using recombinant protein and synthetic peptides, we describe a novel peptidic substrate (Z-EVNL-AMC, JMV4482, Z = benzyloxycarbonyl, AMC = 7-amino-4-methylcoumarin) for HslV and analyze the structural requirements for HslU-derived peptides to activate LmHslV as well as the conformational reorganization of this complex upon activation.

## 2. Results

### 2.1. Expression and Purification of Recombinant LmHslV

In order to characterize *L. major* HslV, we produced the recombinant protein in *E. coli*. As the protease is normally expressed with a N-terminal mitochondrial signal peptide that is removed upon translocation to the mitochondrion, we produced a modified form of the protein, in which the signal peptide has been replaced by a N-terminal methionine that precedes the ‘TTI’ motif required for activity (Appendix A). Indeed, this configuration is that observed in *E. coli* HslV, for which it is known that the N-terminal methionine is cleaved upon expression, thus exposing a N-terminal threonine (Thr1) that is the catalytic residue. Additionally, a C-terminal 6xHis tag was added to the protein for its rapid purification.

After expression in *E. coli*, recombinant LmHslV was easily purified on nickel using a Ni-NTA superflow (Qiagen) column (Appendix A). Depending on the samples, a few high molecular weight contaminants were visible by SDS-PAGE analysis. Therefore, in some experiments we further purified the protein by gel filtration (Superose 12) to resolve it away from the contaminants and obtain purer samples. However, we did not notice any interference of these contaminants with the activity of the protease. As expected, gel filtration analyses showed that the protein was able to spontaneously assemble as a dodecamer (Appendix A), as seen with HslVs from other organisms. As expected, N-terminal sequencing by Edman degradation showed that 89% of the produced recombinant LmHslV polypeptides had the N-terminal methionine removed and exposed an active N-terminal threonine.

### 2.2. LmHslV Is a Dodecameric and Inactive Protease that Can Be Activated by Peptides Corresponding to the C-terminal End of LmHslU1 and U2

#### 2.2.1. Peptides Derived from the C-terminal Ends of LmHslU1 and U2 Can Activate LmHslV

After purification, we initially found no activity for the protease using the Z-GGL-AMC peptide substrate classically used to assess activity of HslV from different species, including *E. coli* [5] and *P. falciparum* [34]. This was most likely due to the absence of HslU, whose binding is known to activate HslV. Unfortunately, we failed to obtain recombinant LmHslU1 and LmHslU2 in a soluble form. Therefore, we tested whether we could activate LmHslV by incubating the complex with peptides derived from the C-terminal end of LmHslU1 or LmHslU2, as previously shown for HslVs from other organisms [15,35].

We first synthesized peptides **1**–**4** corresponding to the 8 C-terminal amino acid residues of LmHslU1 and LmHslU2, acetylated (Ac-) or not (H-) on their N-terminus (H-LmC8-U1, Ac-LmC8-U1, H-LmC8-U2 and Ac-LmC8-U2, respectively, see Table 1 for the sequences of all peptides used in this study) and assessed their effect on LmHslV activity. We also synthesized and tested peptides **6** and **7** corresponding to C-terminal octapeptide of *Escherichia coli*, respectively, non-acetylated (H-EcC8-U) and acetylated (Ac-EcC8-U).

Despite solubility issues at the concentrations in the millimolar range required for this effect, we found that the LmHslU1-derived peptide **1** (H-LmC8-U1) was able to activate the protease, using the substrate Z-GGL-AMC (Figure 1A). Moreover, a robust activation was seen with the acetylated LmHslU2-derived peptide **4** (Ac-LmC8-U2) whereas its non-acetylated analogue **3** was not soluble enough for reliable results. Interestingly, the octapeptides derived from EcHslU, **6** and **7** were also found to activate LmHslV (Figure 1B), in contrast to the results reported by Sung et al. for TbHslV [32].

This first series of experiments demonstrated that LmHslV was virtually inactive in the absence of its activator, and that it could be activated by peptides derived from the C-terminus of various HslUs, including LmHslU1, LmHslU2 and EcHslU. At this stage however, the assays were not robust, due in large part to the poor solubility of the peptides tested.

To improve our tools, we developed a dodecapeptide series since Ramachandran et al. [15] reported that the C-terminal dodecapeptide from *E. coli* HslU (EcC12-U) was about twice as potent as the corresponding octapeptide at activating EcHslV against peptide substrates. For better aqueous solubility, we added a hydrophilic segment composed of a D-arginine and a small PEG segment O2Oc at the N-terminus of the peptides. Preliminary experiments led to the following reference activator peptide derived from LmHslU2, compound **10**: H-arg-O2Oc-Leu^1^-Gln-Lys-Asn-Val-Asn-Leu-Ala-Lys-Tyr-Leu-Leu^12^-OH. This peptide, named LmC12-U2, was found to be more soluble and to activate LmHslV much more efficiently than the LmC8-U2 peptide **4** in the presence of the substrate Z-GGL-AMC.

A further improvement to our assays was the development of a novel fluorogenic peptide substrate, which was more convenient to use than Z-GGL-AMC. Indeed, although the substrate Z-GGL-AMC mostly used in the literature for the studies of HslV has proven useful, its use is limited by its poor solubility in aqueous solutions [6,36]. In view of finding alternative substrates for LmHslV, we tested a variety of possible fluorogenic substrates (commercial or home-made) including Suc-LLVY-AMC, a classical substrate used for the eukaryotic proteasome, in the presence of the reference activator peptide LmC12-U2. As shown in Figure 2, both Z-GGL-AMC and Suc-LLVY-AMC peptides were degraded by peptide-activated LmHslV, as expected from previous work performed on HslV of other species. Most of other tested substrates were either not- or poorly-digested by activated LmHslV. The only exception was the fluorogenic substrate Z-EVNL-AMC (JMV4482), which was in fact more efficiently degraded by LmHslV than Z-GGL-AMC and Suc-LLVY-AMC at 100 μM (Figure 2). Furthermore, we observed no solubility problems at concentrations up to 300 μM whereas Z-GGL-AMC becomes insoluble at concentrations above 100 μM. Interestingly, we observed that Z-EVNL-AMC was also more efficiently degraded by HslV from *E. coli* (EcHslV) than Z-GGL-AMC, suggesting that it could be used to probe activity for HslVs of most species.

Thanks to these improvements, we then undertook a detailed analysis of the sequence and structural requirements for LmHslV activation, using compound **10** (LmC12-U2) as an internal reference in all experiments and Z-EVNL-AMC as a substrate.

#### 2.2.2. Sequence Requirements for Activation of LmHslV by C-terminal HslU Peptides

We tested the ability of dodecapeptides derived from the C-terminal end of LmHslU1 and LmHslU2 and of HslU from other micro-organisms to activate LmHslV.

As shown in Figure 3, the C12-U1 peptide derived from LmHslU1, **8** could readily activate LmHslV, although it was slightly less efficient than the corresponding C12-U2 peptide. As a Tyr residue present in LmC12-U2 but not LmC12-U1 has been identified as critical for HslV activation in *T. brucei* [32], we replaced the corresponding Phe by Tyr in LmC12-U1 (compound **9**). This substitution led to an increase in activity, conversely, replacing Tyr by Phe in LmC12-U2 (compound **11**) led to decreased activity (Figure 3). This result corroborates the conclusion by Sung et al. [32] that this residue determines a different behavior between U1 and U2 C-termini. However, its importance appears less dramatic in *L. major* compared to *T. brucei* since peptides derived from LmHslU1 are able to bind to and activate the protease, contrary to what has been seen in *T. brucei*. Of note, it is unclear at this stage whether the different activation properties of LmC12-U1 and U2 peptides are due to a difference in affinity and/or function between HslU1 and HslU2, or to possible differences in solubility (although C12-U1 was significantly more soluble than C8-U1).

Interestingly, we found that peptides from *T. brucei* (TbC12-U2, **13**) and *P. falciparum* (PfC12-U, **14**) also efficiently activated LmHslV and were twice and 1.5-fold, respectively, as potent as LmC12-U2. In addition, as observed in the octapeptide series, the *E. coli* analogue EcC12-U (**15**) was as potent as LmC12-U2 (Figure 3). These results demonstrate that a given HslV could be activated by HslU C-terminal segments from various sources. They are in agreement with the high conservation of this segment and suggest that a same HslU C-terminal mimetic could represent a general HslVU assembly inhibitor. As expected, a scrambled version of LmC12-U2 (compound **44**) has no activity (Figure 3).

To assess the relative importance of side-chains in its activity, Ala-scanning was performed on the reference compound **10** (LmC12-U2) and concerned the last eight residues of the peptide except, of course, the Ala residue at position 8 (compounds **16**–**22**). Figure 4A shows that five out of the last 6 residues of LmHslU2 C-terminal end are essential for activity (compounds **18**–**22** display very low or no activity). This result is in agreement with similar data obtained with EcC8-U peptide [15] and with the high conservation of these residues among all known HslU sequences. It was hypothesized that Ala^8^ is essential as this residue is not conserved. This was confirmed by the synthesis of cyclic analogues (see below).

Next, *D*-scanning was performed on the last six residues of LmC12-U2, each being replaced by its corresponding *D* isomer (compounds **23**–**28**). Figure 4B shows that replacement of residues 7, 9–12 led to analogues (compounds **23**, **25**–**28**, respectively) with no activity. In the case of Ala8 (compound **24**), a residual activity was observed. In the crystal structure of HslVU of *Haemophilus influenzae* [8], the C-terminal segment of HiHslU forms a short helix involving residues corresponding to the Asp^6^-Tyr^10^ segment of LmC12-U2 and ended by two residues in β-conformation. Therefore, as such modification is known to destabilize α-helices or to significantly change β-conformation, it can be hypothesized that any *D*-residue in this segment does not allow the peptide to display the bioactive conformation. Even though the peptide backbone would have the native conformation, the orientation of the side-chain of the *D*-residue would be different.

We then synthesized peptides progressively shortened from the N-terminus of LmC12-U2 to assess the minimal length required for measurable LmHslV activation. Peptides were tested as two versions: one *N*-terminally acetylated analogue (compounds **4**, **32**, **34**, **36**) and/or one analogue with the hydrophilic segment arg-O2Oc (compounds **29**, **30**, **31**, **33**, **35**). Figure 4C shows that a minimum of six residues is required. As expected, we observed a gradual increase in potency when extending the peptide length, the reference **10**, LmC12-U2, being by far the most potent activator. Accordingly, the analogue with the hydrophilic segment was always found to be slightly more potent than the acetylated counterpart, most likely because it is longer. These results suggest that the N-terminal hexapeptide of LmC12-U2 helps the C-terminal part to acquire the bioactive conformation for HslV binding and activation. This is in agreement with the estimations of the minimum length of peptides required to stabilize a relatively stable α-helix portion in solution [37].

#### 2.2.3. Structural Requirements for Activation of LmHslV by C-terminal LmHslU2 Peptides

As mentioned above, when interacting with HslV, the C-terminal hexapeptide segment of HslU in *H. influenzae* forms a short helical structure, the C-terminal leucine residue being distended from this helix portion [8]. Assuming that a similar conformation should be adopted by every C-terminal HslU segments because of their high sequence conservation, we introduced a conformational constraint aiming at favouring the formation of this helix portion.

It is well known that linking the two side-chains of residues i and i+4 can help to stabilize an α-helix conformation [38]. As Ala^8^ is the only residue considered to be non-essential in the C-terminal hexapeptide segment, this position was chosen as the i+4 position and therefore, i residue was Asn^4^. In addition, the segment i-i+4 contains residues present in the expected helix portion.

As a first choice, we prepared the cyclic compounds **37**–**43** where the side-chains of residues 4 and 8 are linked through a lactam bridge. Molecular modelling based on the crystal structure of *H. influenzae* (pdb 1G3I) confirmed that the bridge moiety of the cyclic peptide **37** points away from the HslV interface, suggesting that this modification should be well tolerated. The two residues involved in the cyclization are aspartyl at position 4 and any diaminoacyl residue at position 8. This offers the advantage to keep the asparaginyl moiety at position 4. During synthesis, these residues were introduced with their side-chains being protected with an allyl and an aloc groups, respectively. After assembly, these two residues were selectively deprotected using phenylsilane (24 equiv.) and Pd(PPh_3_)_4_ (0.2 equiv.) in DCM, and the free carboxylic and amino groups were coupled on the solid phase in the presence of HBTU reagent (Scheme 1).

First, Asn^4^ and Ala^8^ were substituted by an aspartyl and a lysine residue, respectively. Four cyclic peptides were prepared with LL (**37**), DL (**38**), LD (**39**) and DD (**40**) aspartyl/lysyl combinations in order to evaluate the importance of stereochemistry. All four compounds were able to activate the digestion of the fluorogenic substrate by HslV with relative potencies of 55%–85%, the less potent being the DD (**40**) analogue and the most potent being the LL (**37**) one (Figure 5). It is noteworthy that the cyclic LD (**39**) analogue, which contains a *D*-residue at position 8 was much more potent than the linear compound **24**, which also contains a *D*-residue (ala) at the same position. This result suggests that cyclization could compensate a localized conformational disruption caused by the *D*-residue. We assessed the importance of cycle size by making analogues with the LL combinations Asp/Orn (**41**) and Asp/Dap (**42**), decreasing the cycle size by one and three atoms, respectively. As shown on Figure 5, the activity decreased when decreasing the cycle size, the shortest cyclic analogue **42** being almost half as active as the reference peptide.

Therefore, no increase in activity was observed when further constraining the peptide backbone. However, interestingly, when removing the three N-terminal residues of compound **37**, outside the cyclic region, the resulting cyclic [Asp^4^, Lys^8^]-LmC9-U2 compound (**43**) was found to be almost as active as the reference peptide **10,** and therefore, more potent than the corresponding linear LmC9-U2 peptide **29** (Figure 5). This observation indicates that the cycle helped to stabilize the bioactive conformation of the C-terminal region. In agreement with our study of the impact of peptide length, these results support an important role of the N-terminal region in stabilizing the conformation of the C-terminal one.

In conclusion, cyclization allows a decrease in the size of the activating peptide without impairing HslV activation potency.

To assess the effective influence of these modifications on peptide conformation, several compounds (three linear **10**, **29**, **30**, and two cyclic **37**, **43**) were studied by circular dichroism in aqueous buffer with or without trifluoroethanol (TFE), a co-solvent with helix-promoting properties (Table 2 and Appendix A). As expected because of their small size, all peptides mainly display random conformation with very low helix content. Addition of TFE led to a moderate increase of the helicity and the largest variation was affected with 25% TFE, whereas 50% TFE only slightly increased it further. The CD analysis of the linear LmU2 peptides (**10**, LmC12-U2; **29**, LmC9-U2; **30**, LmC8-U2) in the presence of TFE showed that their propensity to adopt α-helix structure was similarly low. The most significant effect was observed with the cyclic peptides. Indeed, the LmC12-U2 analogue **37** showed about 20% helix content at 50% TFE. The C9-U2 analogue **43** showed lower helix content compared to **37**, probably because of its shorter length, but the content was still slightly higher than its linear parent peptide **29**. Overall, the helix content was not clearly related to the HslV activating potency.

Finally, as the interaction of the HslU complex with HslV is multivalent, we explored the effect of LmC12-U2 dimerization. It is well known that oligomeric peptide ligands usually present synergetic affinity toward their receptor by increased avidity [40]. The distance (43.8 Å) on HslV between two adjacent pockets accommodating HslU C-terminal ends was measured using the HiHslV crystal 3D structure (PDB code 1G3I) [8], in order to design an appropriate spacer to link two dodecapeptides. According to this distance, we designed two different monomers, each containing three O2Oc units, with either a 3-azidopropanoyl or a pentynoyl moiety at their N-terminus. The two monomers were then linked together in solution by Huisgen cycloaddition using a copper catalyst (click chemistry) to yield the dimer **12** leading to a 1,2,3-triazole heterocycle (Scheme 2).

The dimer was tested in comparison with the reference peptide **10** under usual conditions. HslV activity in the presence of the dimer was about twice that reached with the monomer (Appendix A). Similar results were obtained for the corresponding dimer of LmC8-U2 **5** (not shown). As the dimer concentration corresponds to double concentration in monomer, these results indicate that dimerization is not enough to induce a synergetic effect.

### 2.3. LmHslV Activation by LmC12-U2 Is Accompanied by a Conformational Rearrangement of the Dodecamer

We analyzed the effect of LmC12-U2 on LmHslV by electron microscopy and cryo-electron microscopy.

Images of negatively stained LmHslV incubated or not with the peptide LmC12-U2 were similar, displaying a uniform field of round shaped particles having an average diameter of ~11–12 nm (Figure 6A). We computed two-dimensional class averages from HslV and HslV-LmC12-U2 particles. They correspond to the average of particles presenting the same orientation, resulting in the increase of the signal to noise ratio and subsequently allowing us to observe the structural features more clearly. In both complexes, top, side and intermediate views were easily distinguishable, as presented in insets Figure 6B,C.

A star-like shape motif presenting six globular domains is visible from the top views, while side views display a rectangular shape structure with vertical and elongated densities. According to an iterative procedure based on the IMAGIC V software and described in the “experimental procedures” section, we computed the three-dimensional reconstructions of HslV and HslV-LmC12-U2 at 24 Å and 26 Å resolution, respectively. LmHslV is organized as a dimer of hexamers presenting a D6 symmetry. When visualizing complexes from side view, HslV subunits from the two opposed hexamers present a small tilt angle with respect to the vertical axis. In the presence of LmC12-U2, HslV complex presents a similar structural organization, though with some obvious structural discrepancies. In particular, the dodecamer appears more elongated and less wide, with the tilt angle between opposed HslV subunits less pronounced, while on the other hand, the central channel appears larger. Although we took care to avoid an over representation of preferential views that could induce some structural distortions in 3D reconstructions, such artifacts could not be totally excluded. Consequently, we conducted a full examination of 2D class averages, revealing that most structural differences observed between 3D reconstructions of HslV and HslV-LmC12-U2 were initially visible in 2D class averages as shown in Figure 7.

We superimposed a circle of 12.5 nm diameter and a rectangle of 12.2 nm × 8.8 nm on the top and side views, respectively. The outer space between HslV-LmC12-U2 subunits and the circle in the top view is more pronounced than for HslV. The centre is heavily stained, indicating the central channel is larger in HslV-LmC12-U2 than in HslV. In side views, HslV is constrained in the rectangle whereas a space between HslV-LmC12-U2 and the lateral edges of the rectangle is unambiguously present. In the orthogonal direction, the white density of HslV-LmC12-U2 goes outside the upper limit of the rectangle indicating a longer distance than HslV.

We also decided to observe frozen-hydrated HslV and HslV-LmC12-U2 complexes in cryo-EM. Appendix A shows a view of HslV-LmC12-U2 complexes. Most of particles display a star-like shape motif representative of top views. Side views are also clearly visible as indicated by white arrows. Cryo-EM maps of HslV and HslV-LmC12-U2 were computed at 25 Å resolution. They are consistent with those previously obtained in negative stain. Consequently, although at low resolution, the structural differences between HslV and HslV-LmC12-U2 were clearly observed, supporting that HslV undergoes global structural reorganization upon LmC12-U2 binding.

## 3. Discussion

Since its first description in bacteria more than 20 years ago, a large amount of information has been collected on the protease HslVU, including the discovery of its existence in primordial eukaryotes. Yet, many aspects of HslVU’s role and mechanism of action are still poorly understood. However, a key mechanism for HslV functioning is its activation upon binding of its partner HslU. Our data provide novel insight on this issue.

### 3.1. Structural Requirements for Activation of LmHslV by C-terminal LmHslU2 Peptides

Since peptides corresponding to the C-terminus of HslU are able to activate HslV active sites [15,32,35], we used peptides derived from the C-termini of LmHslU1 and LmHslU2 to analyze activation of LmHslV. The dodecapeptides that we developed based on previous data [15], N-terminally extended by a hydrophilic moiety, were found to be much better tools than octapeptides for LmHslV activation, probably in part due to better solubility but also because of higher affinity related to a better conformational stability (see below). These dodecapeptides were thus used to study structure-activity relationships.

Ala-scan, D-scan and truncations allowed us to show that as expected from a previous study [15], five out the last six C-terminal residues of LmC12-U2 were essential to bind and activate LmHslV, and that six residues seem to be the minimal length to activate LmHslV (Figure 4). Of note however, the large difference observed between LmC12-U2 and LmC6-U2 indicates that additional *N*-terminal residues have an important role. Knowing that these residues are less conserved in HslUs than the last C-terminal hexapeptides, and that most of them are not in contact with HslV residues within the binding pocket, as shown in HiHslVU crystal structure [8], this role is likely to be more conformational than due to a direct contribution to binding.

As the HiHslU C-terminal segment was shown to be folded as a short helix flanked by residues in the extended conformation when interacting with HslV [8], we hypothesized that a similar conformation should be adopted by every C-terminal HslU segment. To favour the α-helix, we designed several cyclic analogues with a lactam bridge linking the side-chains of an aspartyl and a lysyl residue at positions 4 and 8, respectively. The wild-type L-Asp/L-Lys analogue **37** was found to be the more potent cyclic analogue. Removing the three N-terminal residues outside the cycle (compound **43**) did not reduce the activity compared to LmC12-U2 (**10**) and its cyclic analogue **37**, in contrast to the difference observed between the linear peptides LmC12-U2 (**10**) and LmC9-U2 (**29**). This result confirms the conformational role of the *N*-terminal part of LmC12-U2.

Interestingly, the presence of *D*-Asp and/or *D*-Lys led to compounds (**38**–**40**) with about 60% activity relative to LmC12-U2 **10**. This result is somewhat surprising as the replacement of Ala^8^ by its *D*-enantiomer (compound **24**) led to an 80% decrease in activity, indicating that the cyclisation allowed for partial compensation of the local conformational disturbance and/or steric hindrance caused by the *D*-side substituent.

### 3.2. Specific Roles of HslU1 and HslU2

An intriguing feature of Trypanosomatids is the presence of two isoforms of HslU (HlsU1 and HslU2) in their genome. Evolutionary analyses suggest that HslU1 is more distant in sequence from bacterial HslUs than HslU2 [33]. In addition, it is noteworthy that LmHslU1 and LmHslU2 (sequences accession numbers Q4QFH5 and Q4QIO3, respectively) differ significantly in their I-domain (U1, residues 107–221 and U2, residues 144–303), which is involved in protein substrate recognition and unfolding. This suggests that the two HslU proteins target different types of protein substrates and/or play distinct mechanistic roles in substrate recognition. To clarify this issue, a lingering problem is to understand whether both HslUs are able to bind to and activate HslV, and whether both proteins work together (i.e., in a heterohexamer) or independently (i.e., as homohexamers) in HslV regulation.

In accordance with the hypothesis that both HslUs interact with and activate HslV, in this manuscript we show that peptides corresponding to the C-terminal end of both LmHslU1 (**1** and **8**) and LmHslU2 (**4** and **10**) are able to activate LmHslV (Figure 1 and Figure 3), even though LmHslU1 peptides were systematically less efficient than their LmHslU2 counterpart. However, these observations seem to be in striking contradiction with data obtained with *T. brucei* counterparts showing that although the C-terminal octapeptide of TbHslU2 (TbC8-U2) efficiently activates TbHslV, the corresponding peptide from TbHslU1 (TbC8-U1) cannot do so [32]. The interpretation for the absence of the effect of TbC8-U1 was that it lacks at the antepenultimate position a critical tyrosine, present in TbHlsU2 (Tyr^494^) but replaced by a phenylalanine in TbHslU1 [32]. Indeed, we confirmed an important role for this residue, since we found that replacing Phe by Tyr in LmC12-U1 (**9**) led to a large increase in potency whereas replacing Tyr by Phe in LmC12-U2 (**11**) led to the reverse effect (Figure 3). Since the C-terminal ends of LmHslU1 and TbHslU1 are identical and both lack the critical antepenultimate tyrosine that is present in both LmHslU2 and TbHslU2 (see Appendix A), the simplest hypothesis is that the difference for HslV activation between both organisms resides not in the peptides but in the HslV moiety. The fact that the C-terminal octapeptides from *E. coli* (EcC8-U) can activate LmHslV (this manuscript) but not TbHslV [32] is consistent with this notion, even though a direct comparison between our experiments and those of Sung and colleagues [32] is complicated due to the fact that the peptide concentrations were not indicated in the TbHslV study.

Indeed, a possible explanation for the differential effect of HslU1 peptides in *L. major* and in *T. brucei* could reside in the different structures of the HslU-binding pockets of Lm- and TbHslV. In fact, it has been suggested [32] that the critical role of the antepenultimate tyrosine in TbHslU2 is linked to the fact that TbHslV residues accommodating TbHslU C-termini (namely Ile^54^, Met^57^ and Thr^114^) are smaller than those found in EcHslV or HiHslV (respectively Phe^54^, Phe^57^ and Gln^114^). According to this hypothesis, the presence of the antepenultimate tyrosine in TbHslU could favor tight binding into the large pocket of TbHslV, thanks to a hydrogen bond that can be established between the hydroxyl group and Ala^79^ main chain atoms of TbHslV [32]. Our analysis of the X-ray structure (pdb 1G3I) suggests that the tyrosine could also form a hydrogen bond with Lys^80^ in HslV. Following this rationale, and since LmHslV possesses a phenylalanine at position 54, as EcHslV and HiHslV (see Appendix A), the smaller HslU-binding pocket of LmHslV might not require an antepenultimate tyrosine in LmHslU, explaining why it can accommodate HslU1.

Altogether, our results support the idea that both HslU1 and U2 are functional partners of HslV in trypanosomatids, with the big caveat that it is unclear how much results obtained with C-terminal peptides can be extrapolated to the native, full-length proteins. If it is the case, then an important question is whether they interact with HslV independently to each other or not. The fact that in *T. brucei* the protein HslU1 might not be able to bind to HslV by itself suggests not, but further analyses are necessary to clarify this issue.

Finally, another intriguing conclusion of our work is that the binding of HslUs to HslV is not optimal, indeed, our data indicate that much higher activation of LmHslV can be achieved with HslU C-termini of other organisms than *L. major*, or by mutations of LmHslUs (Figure 3). Most likely it is important for HslVU functioning that the binding between its two sub-complexes is not too tight.

### 3.3. Conformational Rearrangement upon LmHslV Activation

In bacteria, both a widening of its pores [11,12] and an allosteric regulation of its catalytic sites [14,15] seem to cooperate in HslV activation. However, this process remains ill-understood. Our observations showed that LmHslV presents an overall architecture similar to the other previously characterized HslVs. However, EM analyses of HslV structure after both negative staining and cryo-EM reveal unsuspected global structural rearrangements of LmHslV upon activation by LmC12-U2, involving a widening of the central channel and a lengthening and narrowing of the dodecamer accompanied by a slight shift of HslV subunits.

The widening of central pores is consistent with observations related to bacterial HslV activation, but other changes in the overall structure of HslV were not observed compared to crystal structures of free and HslU-bound (i.e., activated) bacterial HslVs [8,11,14,41,42]. However, it is interesting to note that a “twist-and-open” mechanism has been described for bacterial HslV upon HslU binding [9]. This twist of the HslV ring associated to a widening of its pores somehow resembles, albeit at a smaller scale, the structural reorganization observed for LmHslV. It is thus possible that LmHslV evolved by amplifying the initial conformational rearrangement already present in its bacterial ancestor, possibly as a mechanism to tighten the regulation of its activity.

An intriguing question is how such a reorganization can occur without ATPase activity in a LmHslV dodecamer, which has a structure that is quite constrained. Crystal structures of latent and activated LmHslV will most likely be necessary to solve this question. It is already established that upon HslU binding, the HslV structure is dynamic enough to undergo both a reorganization of the pores and conformational changes that are propagated up to the active sites [43]. Thus, we can suppose that there is a plasticity in the manner that HslV subunits are assembled that might eventually authorize major conformational changes at a small energetic cost.

### 3.4. Drugging HslVU

Because of its essential role in trypanosomatids and absence in humans, the HslVU complex represents a promising target to extend the presently limited drug arsenal to fight against these deadly parasites. As the catalytic mechanism of HslV is shared with the proteasome, developing drugs specifically targeting HslV active sites without affecting the host proteasome is extremely challenging. In this context, one alternative and relevant approach is to develop compounds targeting HslVU assembly instead of its catalytic sites. One evident angle of attack is the interaction of the HslU C-terminal segment with HslV, which is involved in both complex assembly and HslV activation, and to design molecules that will prevent HslU association by occupying its binding pockets on HslV. Our present data show that there is room to develop compounds that have much more affinity for these pockets than HslUs. Furthermore, in principle there is the possibility to multimerize such molecules to increase avidity and efficiently compete with HslU, even though our present data indicate that dimerization might not be sufficient for such a synergistic effect. It is obvious that the size of the molecules, as well as their targeting of the parasite mitochondrion, are the main constraints that might limit their development into drugs that could be used in patients. However, although the longest peptides tested in this study (i.e., dodecapeptides) were by far more potent than the shorter ones, we were able to decrease the peptide length without a decrease in activity by conformational stabilization. Our results make us confident that even shorter peptides or peptidomimetics with similar or enhanced activity will be discovered through further careful exploration of structure-activity relationships. These studies are ongoing in our laboratories.

Independently of its future development, our present approach shows the value of using peptides to probe the structural requirements allowing strong binding into the HslU-binding pockets of HslV. Furthermore, an unexpected result of our peptide-based approach was to show that their binding energy was enough to support an important conformational shift of the protease. Interestingly, this conformational reorganization might be a novel entry point towards specific inhibition of parasitic HslVs, as one might imagine blocking it, thus, preventing HslV activation with small molecules binding into regions critical for HslV reorganization.

## 4. Materials and Methods

### 4.1. Material

All peptides (Table 1 and Appendix A) were assembled on a 2-chloro-trityl resin following either manual or microwave-assisted solid phase synthetic protocols with Fmoc as the Nα-protecting group and HBTU or HATU/DIEA as coupling agents. Their synthesis as well that of the substrate JMV4482 (Z-EVNL-AMC) routinely used for activity tests are described in detail in the Appendix A. Suc-LLVY-AMC and Z-GGL-AMC were purchased from Bachem. pET-Duet-1 plasmid with inducible *E. coli* C-terminally 6His-tagged HslV were kindly given by Dr Chin Ha Chung from the Department of Biological Sciences of Seoul National University.

### 4.2. Methods

Cloning of LmHslV: LmHslV gene was amplified with its mitochondrial signal sequence using DNA from *Leishmania major* Friedlin strain and subcloned into PQE60 vector to obtain the C-terminally 6xHis-tagged LmHslV protein. For expression in *E. coli*, LmHslV-6His was subcloned without its mitochondrial targeting signal into the vector pRSET-B.

Protein expression: The proteins of interest (*L. major* and *E. coli* HslV, both with a C-terminal 6xHis tag), were expressed overnight at 20 °C in *E. coli* BL21 (LB medium + ampicilline 100 μg/mL), by adding 1 mM IPTG in a culture with an OD between 0.5 and 0.8.

Protein purification by Ni^2+^-NTA chromatography*:* All purification steps were performed at 4 °C. Bacteria cultures (usually 500 mL) were centrifuged at 4000 RPM for 15 min, the pellet was collected, washed with PBS, then resuspended in 45 mL lysis buffer (PBS 2X, 10 mM Tris-HCl pH 8.0, 1 mM EDTA). Bacteria were then lysed with a high-pressure homogeneizer (Emulsiflex-C3, Avestin, Ottawa, Canada). 5 mL of 10% Triton X-100 (1% final) was added to the homogenate, which was then clarified by centrifugation (1,2000× *g*, 20 min). The supernatant was collected and incubated overnight with 500 μL of Ni-NTA beads (Qiagen, les Ulis, France) pre-equilibrated with PBS 2X. After their collection, the Ni-NTA beads were successively washed (several column volumes) with (1) PBS 2×, 1% Triton X-100, (2) PBS 2×, 30 mM imidazole and (3) Tris-HCl 20 mM pH 8.0, 30 mM imidazole. Elution (2 mL) was performed with 250 mM Imidazole in 20 mM Tris-HCl pH 8.0. Purified proteins were then dialyzed overnight against 20 mM Tris-HCl pH 8.0, 100 mM NaCl, 0.5 mM EDTA, 10% Glycerol, 5 mM MgCl_2_ and 1 mM DTT (buffer A).

Analytical gel filtration analysis of recombinant LmHslV: LmHslV (10 µg) was loaded on a gel filtration column (Superose 12 PC 3.2/30) equilibrated with buffer A; 15 fractions of 100 µL were collected.

Activity assay: The peptidase activity of recombinant LmHslV was assessed by measuring the fluorescence of AMC released after hydrolysis of the fluorogenic substrates Z-GGL-AMC or JMV4482 (Z-EVNL-AMC), 100 µM. LmHslV activation was obtained by addition of activating peptides (see type and concentration in the figure legends). Activity buffer: 25 mM Tris-HCl pH 8.0, 1 mM DTT, 0.5 mM EDTA and 5 mM MgCl_2_. Most of the experiments were performed in a final volume of 50 µL in black 96-well microplates (Fisher Scientific, Illkirch, France. Fluorescence was measured using a spectrofluorimeter (BIO-TEK Instrument, Inc., Winooski, VT, USA): filters 360/340 for excitation and 460/440 for emission.

Electron microscopy: In this study, LmHSLV samples were observed both by TEM after negative staining and by cryo-EM. LmHslV was diluted at 150 nM (final concentration) in 25mM Tris-HCl pH 8.0, 0.5 mM EDTA, 5mM MgCl_2_, 1 mM DTT, in the presence or absence of 500 μM of the activating peptide LmC12-U2. Samples were then preincubated during 30 min at 37 °C. For negatively stained samples, three microliters of samples at a final concentration ranging from 0.05 to 0.1 mg/mL were applied for 2 min to glow discharged carbon-coated grids, and staining with 1% uranyl acetate for 1 min. For cryo-EM experiments, three microliters of LmHSLV samples at ~1–2 mg/mL were applied to glow-discharged Quantifoil R 2/2 grids (Quantifoil Micro tools GmbH, Grossloebichau, Germany), blotted for 1s and then flash frozen in liquid ethane using the semi-automated plunge freezing device CP3 (Gatan inc.) at 95% relative humidity. Images were recorded on a JEOL 2200FS FEG operating at 200 kV equipped with a 4 k × 4 k slow-scan CDD camera (Gatan Inc., Pleasanton, CA, USA) under low-dose conditions (total dose of 20 electrons/Å^2^) in the zero-energy-loss mode with a slit width of 20 eV. Images were taken at a nominal magnification of 50,000 × and 100,000 × for negative staining and cryo-EM, respectively, with defocus values ranging from 0.6 to 2.5 μm. Magnifications were calibrated from cryo-images of tobacco mosaic viruses.

Image processing: Images were checked for drift or astigmatism and particles extracted semi-automatically using e2evalimage and e2boxer, respectively, from the Eman2 package [44]. All next steps of image processing were performed using IMAGIC V software [45] according to the method presented in Bron et al. [46]. Briefly, the phase-contrast-transfer function was corrected by phase flipping using defocus parameters obtained using Gctf [47]. As the initial iterative alignments of negatively stained LmHslV particles revealed common structural features with other already characterized HslV, particles were then consequently aligned iteratively against the envelop of bacterial HslV (pdb 1NED) filtered at 40 Å using the multi-reference alignment program. Images were then grouped into classes and averaged using the MSA (multi-statistical alignment) procedure. The best class averages were then used as references to perform a new alignment cycle of particle images. Basically, two or three alignment cycles were performed. The best class averages covering all Euler orientations were then selected in avoiding over representation of preferential views and used to compute a three-dimensional model by angular reconstitution with the envelop of bacterial HslV (pdb 1NED) filtered at 40 Å as a reference. The three-dimensional structures were iteratively refined in comparing the reprojections of the three-dimensional models with their corresponding class averages. The best results were obtained when applying a D6 symmetry. The resolution of maps was estimated using an FSC (Fourier shell correlation) coefficient of 0.5 [48]. Surface representations of EM-maps were performed using Chimera software [49].

Sequence alignments: All sequence alignments were performed using the ClustalW software (http://embnet.vital-it.ch/software/ClustalW.html).

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
