# Peer review of "The HslV Protease from Leishmania major and Its Activation by C-terminal HslU Peptides"

_ijms, 2019, doi:10.3390/ijms20051021_

Round 1
Reviewer 1 Report
The manuscript by Coux and colleagues describes the design and assay of a range of short peptides to activate HslV protease complex from the Leishmania parasite. The introduction presents a comprehensive basis for the manuscript, and the list of tested peptides sufficiently probes the various properties required for high activation (e.g. length, specific sidechains, sequence). The top peptide is also shown to alter molecular details of the HslV dodecamer by electron microscopy.
The experiments appear to support the main findings, but several concerns are related to aspects of the results section, primarily section 2.2:
- the presentation of the peptides in Table 1 is difficult to read and unnecessarily detailed with respect to what is required to understand the results. It is highly suggested to reduce the table in the main text to simply the compound number, name and structure (one line per peptide), and optimally to fit on one page. The mass, m/z and LC-MS are not required in the results section and can comprise a new table in the Materials and Methods (or alternately as a Supplemetary Table)
- the order of the first peptides (line151-158) should match the order of the peptides in Fig. 1a
- the compound numbers should be added to the x-axis peptide names in Fig. 1
- it is not clear why Fig. 1b shows relative activity instead of the arbitrary units in Fig. 1a. This type of relative comparison makes more sense with a series of mutants such as in Fig. 4 as compared to the parent peptide, but it less valid for a comparison of unrelated peptides. All of the tested peptides in Fig. 1 should be presented in the same histogram figure (instead of two separate figures)
- the transition from the initial 4 peptides and initial substrate to the new compound and new substrate (lines 166-174) makes it unclear why the initial experiments were performed. There should ideally be something in common between the two assays (such as new substrate with initial peptides, or the new 12mer peptides with the initial substrate. Otherwise the initial assays in Fig. 1 has no comparable link
- the new substrate also lists activating peptides (such as 30) from later in the manuscript which makes it difficult to follow
- the usefulness to other species HslV (lines 188-190) should be better placed in the discussion, especially since the data is not shown
- In Fig. 3 it is again not clear why compound 10 is set as the activity reference for this series
- the analysis of the competition can perhaps be deleted since the data is not included and as such does not significantly add to the results
- the attempt to study enhanced helicity is lacking some supporting data. It is not clear from the results that a helix is in fact significantly stabilized by the series of cyclic peptides, especially since the optimal conditions (TFE) only slightly alter the CD spectra. It is also not clear why the D spectra were not collected first in order to test the peptides with the most stabilized helical content (and not the other way around). That way any activity trends could be better interpreted. Similarly, the table should also include the CD determined helicity (reading at 222 nm) as well as the deduced helix %.
- it would be interesting to have the authors comment on the possibility of testing the earlier peptides on HslV but in the presence of TFE (to increase the helicit) and see if this alone has a positive effect (vs HslV alone in the presence of TFE)
- in the section from lines 334-338 it is not sufficient to state that the data is not shown. Since simple numerical values are enough to judge the relative activities, these numbers should be explicitly stated in place of the 'data not shown'
- there is only one comment or question for section 2.3, and that is whether ATP addition (without or with peptide) was tested as well, in order to look for changes in the molecular shape of the HslV dodecamer
- in the discussion, the PDB of the HslUV complex is mentioned, but perhaps this structure can be better exploited to provide some possible explanations for the main findings of the paper. This could include the potential direction of the best cyclic peptide (for example, does the bridge moiety point away from the HslV interface and that is also perhaps why it is the best tolerated?)
Author Response
The manuscript by Coux and colleagues describes the design and assay of a range of short peptides to activate HslV protease complex from the Leishmania parasite. The introduction presents a comprehensive basis for the manuscript, and the list of tested peptides sufficiently probes the various properties required for high activation (e.g. length, specific sidechains, sequence). The top peptide is also shown to alter molecular details of the HslV dodecamer by electron microscopy.
The experiments appear to support the main findings, but several concerns are related to aspects of the results section, primarily section 2.2:
- the presentation of the peptides in Table 1 is difficult to read and unnecessarily detailed with respect to what is required to understand the results. It is highly suggested to reduce the table in the main text to simply the compound number, name and structure (one line per peptide), and optimally to fit on one page. The mass, m/z and LC-MS are not required in the results section and can comprise a new table in the Materials and Methods (or alternately as a Supplemetary Table)
Table 1 has been modified accordingly and a new table (Table S1) containing MS and HPLC data has been added in Supplementary Information. Consequently the former Table S1 is now Table S2.
- the order of the first peptides (line151-158) should match the order of the peptides in Fig. 1a
The figure has been modified as requested. This modification led us to modify the corresponding text for clarity (lines 146-148 and 151-159), without changing the overall message conveyed by the initial text.
- the compound numbers should be added to the x-axis peptide names in Fig. 1
Done
- it is not clear why Fig. 1b shows relative activity instead of the arbitrary units in Fig. 1a. This type of relative comparison makes more sense with a series of mutants such as in Fig. 4 as compared to the parent peptide, but it less valid for a comparison of unrelated peptides. All of the tested peptides in Fig. 1 should be presented in the same histogram figure (instead of two separate figures)
Both panels show now activity in arbitrary units. However, we prefer to present the data in 2 distinct panels as the experiments were not performed at the same time. To allow comparisons between the 2 experiments, we introduced in each of them an assay using the peptide 4.
- the transition from the initial 4 peptides and initial substrate to the new compound and new substrate (lines 166-174) makes it unclear why the initial experiments were performed. There should ideally be something in common between the two assays (such as new substrate with initial peptides, or the new 12mer peptides with the initial substrate. Otherwise the initial assays in Fig. 1 has no comparable link
We agree to this comment. To clarify the logic of the data presentation, we now explain more precisely the technical difficulties encountered with the octapeptides (lines 163-166), and the developments that we made to improve our assay (see in particular lines 176-177, 186-188, 212-215).
In these pages, Figure 2 has been moved below its initial position to be better integrated within the text.
- the new substrate also lists activating peptides (such as 30) from later in the manuscript which makes it difficult to follow
Compound 30 is an arg-O2Oc-containing version of LmC8-U2 and was synthesized for the study of truncated peptides. As we understand that it could be confusing and that it is not essential to be cited there, we removed “and 30 (see below)” line 185.
- the usefulness to other species HslV (lines 188-190) should be better placed in the discussion, especially since the data is not shown
This comment refers to a sentence appearing now lines 200-202 in the reorganized text. Although we agree with the comment, we thought that it would be hard to place the sentence in the discussion, knowing that the discussion does not address the question of this novel substrate. We thus believe that it is simpler to leave the sentence at its initial position.
- In Fig. 3 it is again not clear why compound 10 is set as the activity reference for this series
We thought that having an internal positive control assay as a reference in each experiment will allow comparing the results obtained with different peptides more easily, even when the experiments are made at distant times and with different HslV samples. Compound 10 (LmC12-U2) was chosen as the reference activator dodecapeptide because it was found more potent than LmC12-U1, and was soluble at the concentrations used. Having this internal reference was important to study the structural requirements for HslV activation, analyses that were therefore performed on the LmC12-U2 basis. At the end of section 2.2.1, we added the following sentence: “Thanks to these improvements, we then undertook a detailed analysis of the sequence and structural requirements for LmHslV activation, using compound 10 (LmC12-U2) as an internal reference in all experiments, and Z-EVNL-AMC as a substrate.”
- the analysis of the competition can perhaps be deleted since the data is not included and as such does not significantly add to the results
We agree to this comment and consequently removed this part (lines 256-260).
- the attempt to study enhanced helicity is lacking some supporting data. It is not clear from the results that a helix is in fact significantly stabilized by the series of cyclic peptides, especially since the optimal conditions (TFE) only slightly alter the CD spectra. It is also not clear why the D spectra were not collected first in order to test the peptides with the most stabilized helical content (and not the other way around). That way any activity trends could be better interpreted. Similarly, the table should also include the CD determined helicity (reading at 222 nm) as well as the deduced helix %.
The short peptide sequences of LmC12-U2 and LmC9-U2 were cyclized to try to stabilize a single helix turn presumably present in the C-terminal structure of LmHslU2 according to the X-ray structure of HiHslVU. Indeed, such feature could increase the activity of the parent peptides and/or stabilize a bioactive conformation. No increase of activity was observed between linear and cyclic dodecapeptides, but a significant increase was found for the cyclic nonapeptide compared to the linear analogue, suggesting a better conformational stability favorable to activity. However, no peptide showed a higher potency than the LmC12-U2 reference peptide. To try to establish a potential link between conformation and measured activity, we performed a CD study of a limited number of analogues in the presence of TFE. Slight but significant conformational change was observed for the cyclic peptide 37, which was not found more potent than 10. Such change has been hardly seen for the cyclic nonapeptide 43, which, in contrast, is more potent than the linear analogue 29, suggesting that the resulting constraint was indeed favorable for the nonapeptide analogue but neutral for the dodecapeptide, but with no clear relation with helix content.
In fact, our objective was to stabilize a potential helical turn by introducing the cyclic constraint in the C-terminal hexapeptide, but not to obtain a fully helical peptide as this conformation is not present in the HiHslVU structure, suggesting that peptides with high helicity could be inactive. There was therefore no reason for us to check CD profiles first. In addition, it was necessary to first assess the structural requirements for HslV activation to know where to introduce the cyclic constraint. Finally, in our opinion, to test only the peptides with the higher helical content could be detrimental since the structure-activity relationship is not clearly demonstrated.
The text has been modified on lines 293-294, 298, 340, 350-352 of the new version to better explain our approach. We also concluded the reporting of this study by: “Overall, the helix content could not be clearly related to the HslV activating potency” (lines 351-352).
Table 2 has been modified accordingly.
- it would be interesting to have the authors comment on the possibility of testing the earlier peptides on HslV but in the presence of TFE (to increase the helicit) and see if this alone has a positive effect (vs HslV alone in the presence of TFE)
It is not clear what could be the effect of TFE on the HslV protease activity. Even if the protease could be still working in these conditions, it would be impossible to distinguish TFE conformational effects on activator peptide and on protease. We therefore did not perform such experiments.
- in the section from lines 334-338 it is not sufficient to state that the data is not shown. Since simple numerical values are enough to judge the relative activities, these numbers should be explicitly stated in place of the 'data not shown'
This refers to lines 367-371 in the actual version. To answer this concern, we added a novel figure (Fig. S3) in the supplementary material section.
- there is only one comment or question for section 2.3, and that is whether ATP addition (without or with peptide) was tested as well, in order to look for changes in the molecular shape of the HslV dodecamer
ATP is required for HslU, but not HslV activity. Since we could not produce and work with LmHslUs, we performed all our assays on HslV only and thus without ATP. Therefore our results clearly show that the binding energy of the peptides is sufficient to induce a global structural rearrangement of HslV. Whether ATP-binding to the protease could have an effect on its conformation remains an open question, although there is to our knowledge no clear reason to expect such a result.
- in the discussion, the PDB of the HslUV complex is mentioned, but perhaps this structure can be better exploited to provide some possible explanations for the main findings of the paper. This could include the potential direction of the best cyclic peptide (for example, does the bridge moiety point away from the HslV interface and that is also perhaps why it is the best tolerated?)
We checked the position of the bridge of compound 37 when bound to HslV and found that it was indeed pointing away from the binding site, explaining why it was tolerated by the protease. We added this information line 301.
We also used the structure to better understand the role of the antepenultimate Tyr in TbHslU2 and LmHslU2 within the HslU-binding pocket of HslV. We modified lines 497-504 to clarify the discussion on this issue and to address the second comment reviewer 2 (see below).
Reviewer 2 Report
@page { size: 8.5in 11in; margin: 0.79in } p { margin-bottom: 0.1in; line-height: 115%; background: transparent }My questions concern the structural aspects of the studies.
1. In Figure S4A authors should highlighted (for instance by red box) the C-terminal Tyr and Phe in HslU1 and HslU2, respectively. In Figure S4B the sequence of Haemophilus influenzae have to be include in alignments of HslV’s. Moreover, the sequence of HiHslV is presented in the caption (Uniprot P43772), but not presented in the figure.
2. Authors should add few sentences about structural aspects of HslUV complex? At least, authors should discuss the fact that peptides with Tyr in position 10 bind more tightly compare to Phe one. Analysis of X-ray structure (pdb 1G3I) shows possibility create (at least in HiHslUV complex) hydrogen bond between Tyr in HslU and Lys80 in HslV.
Minor point. I couldn’t understand this phrase (page 15): “Cryo-EM maps of HslV and HslV-LmC12-U2 were computed with final resolutions similar to those obtained in negative stain and according to the same protocol”. Which exact resolution was reached with refine Cryo-EM?
Author Response
My questions concern the structural aspects of the studies.
1. In Figure S4A authors should highlighted (for instance by red box) the C-terminal Tyr and Phe in HslU1 and HslU2, respectively. In Figure S4B the sequence of Haemophilus influenzae have to be include in alignments of HslV’s. Moreover, the sequence of HiHslV is presented in the caption (Uniprot P43772), but not presented in the figure.
These modifications have been done. Please note that Fig S4 is now Fig S5.
2. Authors should add few sentences about structural aspects of HslUV complex? At least, authors should discuss the fact that peptides with Tyr in position 10 bind more tightly compare to Phe one. Analysis of X-ray structure (pdb 1G3I) shows possibility create (at least in HiHslUV complex) hydrogen bond between Tyr in HslU and Lys80 in HslV.
As mentioned above in our answer to the last comment of reviewer 1, we indeed added more details regarding the role of Tyr10 (lines 497-504). In the Sung et al article, the authors mention that Tyr10 could make a hydrogen bond with Ala79 of HslV. As suggested by the reviewer, we rather believe that Tyr10 creates a hydrogen bond with Lys80. This information has been added to the text.
Minor point. I couldn’t understand this phrase (page 15): “Cryo-EM maps of HslV and HslV-LmC12-U2 were computed with final resolutions similar to those obtained in negative stain and according to the same protocol”. Which exact resolution was reached with refine Cryo-EM?
This sentence has been rephrased as follows (lines 421-423): “Cryo-EM maps of HslV and HslV-LmC12-U2 were computed at 25 Å resolution. They are consistent with those previously obtained in negative stain.”